# The Effect of Blocker Application on Color Matching of Different Colored Composite Resin Systems

**DOI:** 10.3390/ma16144954

**Published:** 2023-07-12

**Authors:** Emine Atasayar, Nuran Ulusoy

**Affiliations:** Department of Restorative Dentistry, Faculty of Dentistry, Near East University, Mersin 10, Nicosia 99138, Türkiye; nuran.ulusoy@neu.edu.tr

**Keywords:** color, color matching, blocker, composite resin

## Abstract

To evaluate the effect of blocker on color matching of different shaded composite systems on natural teeth, 120 anterior teeth were divided into two groups as light and dark shades (n = 60). Round cavities (7 mm in diameter; 2 mm in depth) were prepared on labial surfaces of the teeth. They were restored using three types of resin composites: multi-shaded (Estelite Sigma Quick, Tokuyama Dental, Japan), single-shaded (Omnichroma, Tokuyama Dental, Japan) and group-shaded (Neo Spectra ST, Dentsply Sirona, Germany) (n = 20). Blocker (Omnichroma) was applied to half of the samples in each group (n = 10). Color matching of the restorations with the surrounding tissues were evaluated either by visual analysis or an instrumental method. The resin composites tested did not yield acceptable results instrumentally. Using blocker with single-shade resin composite on dark-colored teeth yielded a minimal change in color in 2 mm deep cavities in the visual evaluation by dentists. Similarly, applying blocker with group-shaded and multi-shaded resin composite on light-colored teeth caused minimal change in color of 2 mm deep cavities in the visual evaluation by dentists. As the application of blocker had no effect on the color matching of different shaded resin composites in the visual evaluations by all observers, except by dentists, blocker may be used as a dentin shade in 2 mm deep cavities of anterior teeth.

## 1. Introduction

During the restoration of missing tooth tissues, the aesthetic demands of patients are important as well as the form and function of the teeth. It is essential to have a color match between the color of the teeth and restorative material to achieve an aesthetic result. The color selection becomes more difficult if the natural teeth have a polychromatic structure. Various enamel and dentin shades of resin composites have different opacity and translucency according to the VITA Classical color guide. Consequently, color selection and resin composite application are very sensitive and complex, leading to increasing the chair time in clinics [1,2]. The “blending effect” (BE) of a material refers to its ability to acquire a color similar to that of the surrounding tissue of the teeth. The blending effect in dental restorations is primarily perceived by individuals and cannot be directly quantified using conventional colorimetric devices. However, the translucency of dental materials is believed to contribute to the overall visual acceptability of restorations [3] The blending effect of resin composites with modified optical features has resulted in a decreased range of color shades [4,5]. There are two methods to evaluate color: the instrumental method and the visual method [6]. Instrumental analysis has been conducted using a spectrophotometer, colorimeter, spectroradiometer or digital camera since the 1970s [7]. Spectrophotometers that have standard lighting (6500 K) are most commonly used in the instrumental method.

Spectrophotometers are devices used to measure the amount of light reflected from an object and convert it into measurable data. The Vita Easy Shade Compact (VITA Zahnfabrik Bäd Säckingen, Germany) is a spectrophotometer commonly used in dentistry. It illuminates the tooth with 6500 K light for color matching and provides results in L *, a * and b * values. Various formulas have been used to evaluate color matching instrumentally. While the CIELAB formula has been commonly used for color matching of dental restorative materials, the CIEDE2000 formula is currently the most approved method for evaluating ΔE_00_ (color difference) [8,9]. Visual and instrumental evaluations play a crucial role in aiding in the selection and assessment of dental materials, as well as the interpretation of findings in clinical dentistry and research. The perception of color differences and their acceptability is of utmost importance. Dental research has utilized the comparison between visual and instrumental hue matching to establish dental hue guidelines and explore aspects such as color compatibility, stability and interaction [8]. Visual color selection is the most common method used in dentistry, which involves comparing the standard color of the tooth with tooth color shade guides [1]. In the literature, visual evaluation methods have shown better results than spectrophotometers when comparing visual and instrumental color matching methods [10]. There are various methods in the literature for analyzing color matching, which are necessary for determining clinical and/or perceptibility and acceptability parameters used in color difference analysis (ΔE) [11,12] Perceptibility refers to the color difference between a tooth and a restoration, while acceptability refers to the visual color acceptability of that tooth [8,9]. Perceptibility and acceptability indexes were defined as 0.8 and 1.8, respectively [8].

The purpose of this study was to evaluate and compare the effect of blocker application on the color matching of light and darker colored natural teeth restored with single-shaded, group-shaded and multi-shaded resin composites. Both visual and instrumental methods were used for the evaluation. The study hypotheses are as follows:
**Hypothesis** **1.***There is no difference in color matching among single-shaded, group-shaded and multi-shaded resin composites.*
**Hypothesis** **2.***The color of the tooth (light/dark) does not have an effect on the color matching of the tested resin composites.*
**Hypothesis** **3.***Blocker application will have a positive effect on the color matching between resin composites and natural teeth in both visual and instrumental evaluations.*

## 2. Materials and Methods

Ethical approval (YDU/2022/108-1655) was granted by the Near East University Scientific Research Ethics Committee on 30 November 2022, ensuring compliance with ethical standards and participant protection. In the first method of this study, a spectrophotometer (VITA Easy Shade Compact, VITA Zahnfabrik, Bäd Sackingen, Germany) was used to determine the effect of the blocker on the color matching of three different resin composites with natural teeth. This instrumental method involved measuring and analyzing the color data. In the second method, the color matching was assessed visually, using a grading system, by both dentist and non-dentist observers. These observers visually evaluated and compared the color matching of the resin composites with the natural teeth.

### 2.1. Tooth Preparation

The teeth used in the study underwent a thorough examination, and those with cracks, decay and restorations were excluded. Prior to the commencement of the cavity preparation, the teeth were stored in distilled water. Round-shaped cavities measuring 7 mm in diameter and 2 mm in depth were created on the buccal surfaces of the test teeth using a fissure bur. The standards of the cavities were checked using a digital compass. Following the manufacturer’s instructions, an adhesive (Primer and Bond Universal, Dentsply Sirona, Bensheim, Germany) was applied to the cavities. Half of the teeth in each group (n = 10) were treated with blocker resin composites (Tokuyama Dental, Tokyo, Japan), and the teeth were restored using three different composite materials. Incremental layering technique was employed for each application, with the layers being 1 mm thick. The resin composite restorations were polymerized for 20 s at an energy level of 850–1000 mW/cm^2^ using a light device (Woodpecker LED B; Guilin Woodpecker Medical Instrument, Guilin, China). Following the completion of the restorations, a four-disc grit sequence (Super–Snap Rainbow Kit Shofu Inc., Kyoto, Japan) was applied with water, utilizing a rotational hand tool on low speed. A one-step finisher and polisher (One Gloss Shofu Inc., Kyoto, Japan) were used for the final polishing. Color measurements were performed after the teeth had been stored in distilled water at 37 °C for 24 h.

### 2.2. Instrumental Evaluation

A total of 120 anterior natural teeth were categorized into two groups based on their color: lighter colored (A1, B1, C1, A2, B2) and dark colored (A3, A3.5, B3, B4, C3), using a spectrophotometer [13]. Within each group, the teeth were further divided into three sub-groups for single-shaded, group-shaded and multi-shaded resin composite application. Additionally, the three groups were split into two sub-groups, one with blocker and one without blocker (n = 10). The groups are depicted in Figure 1.

“Omnichroma” (Tokuyama Dental, Tokyo, Japan) is a single-shaded composite designed to harmonize with tooth colors ranging from A1 to D4. “Blocker” (Tokuyama Dental, Tokyo, Japan) is intended for use as a thin layer in the lingual cavity walls of Class I, III, IV and V restorations, serving to mask mild color changes or enhance opacity. “Neo Spectra ST” (Dentsply Sirona, Bensheim, Germany) comprises five different “cloud shades”, each corresponding to a specific VITA color shade group. This system enables color matching with all shades on the VITA scale. Cloud shade A1 was used for tooth shades A1, B1 and C1, while cloud shade A2 was employed for A2 and B2 tooth shades. Cloud shade A3 was selected for A3 tooth shades and A3.5 for B3, B4 and C3 tooth shades. “Estelite Sigma Quick” (Tokuyama Dental, Tokyo, Japan) is a multi-shaded resin composite that includes all existing enamel shades.

The composite materials used in this study are listed in Table 1. Baseline color parameters (L*, a* and b* values) were measured using a spectrophotometer (VITA Easyshade Compact, VITA Zahnfabrik, Bäd Säckingen, Germany) on the flat buccal surfaces of the unrestored teeth. Measurements were taken with a neutral grey background, and the spectrophotometer was recalibrated after every three measurements. The instrumental measurements of teeth were conducted under D65 lighting conditions in a dental clinic room without windows. Three color measurements were taken for each tooth and composite restoration, and the average values were calculated.

The CIEDE2000 color difference (ΔE_00_) was calculated using an electronic table application in Excel, employing the CIEDE2000 color difference formula provided by Lou, Cui and Rigg [14]. The parametric factors of the formula were adjusted to 2.1.1 [9]. 

### 2.3. Visual Evaluation

Visual color evaluations were conducted by a group of participants consisting of five PhD students in dentistry and five persons who were not dentists. Prior to the evaluations, all participants demonstrated proficiency in color discrimination as per the Ishihara color blindness test, in compliance with ISO/TR 28642:2016 [15]. Detailed instructions regarding the evaluation process were provided to the observers.

The evaluations took place under D65 lighting conditions, utilizing 0°/45° screening geometry. Participants visually assessed the samples without knowledge of the materials or color groups. A neutral grey background was employed during the sample evaluation. Each participant was allotted 25 s to evaluate each sample.

Participants were instructed to grade the color match between each tooth and its restoration. The color differences between each tooth and restoration were graded between 0 and 4; 0—complete full up/no difference, 1—very good match up/small difference, 2—good match up/acceptable, 3—bad match up/not very acceptable, 4—disharmony/completely unacceptable.

### 2.4. Statistical Analysis

A power analysis was conducted using analysis software (G*Power, Version “3.1.9.2”) to determine the necessary sample size. The sample size for each group was determined to be a minimum of 5, with α = 0.05, 80% power and a 95% confidence level. To ensure the desired statistical power, 10 samples were prepared for each test group. IBM SPSS 25 software was utilized for the statistical analysis. The normality of the data distribution was assessed using Shapiro–Wilk tests. A trilateral variant analysis test was employed to compare the parameters among the groups. Post hoc Bonferroni test was conducted to identify the specific groups that exhibited significant differences. Statistical significance was accepted as *p* < 0.05.

## 3. Results

The instrumental ΔE_00_ analysis results are presented in Table 2. The usage of Omnichroma with blocker on light-colored teeth (OB-L), Omnichroma with blocker on dark-colored teeth (OB-D), Omnichroma without blocker on light-colored teeth (O-L) and Omnichroma without blocker on dark-colored teeth (O-D) resulted in the highest ΔE_00_ values, and the difference was found to be statistically significant (*p* < 0.05). ΔE_00_ values for all the other groups ranged from 2.64 ± 1.00 to 3.79 ± 1.20, with no significant statistical differences between them (*p* > 0.05). The average values of the visual evaluation are provided in Table 3. Among all observers, the usage of Estelite Sigma Quick without blocker on dark-colored teeth (E-D; 1.49 ± 0.59) achieved the highest scores. However, there were no statistical differences observed in the remaining groups. Dentists specifically noted that blocker application resulted in minimal changes in color matching for the usage of Estelite Sigma Quick with blocker on light-colored teeth (EB-L; 1.24 ± 0.82), Neo Spectra ST with blocker on light-colored teeth (NB-L; 1.78 ± 0.69) and Omnichroma with blocker on dark-colored teeth (OB-D; 1.32 ± 0.61). The dentists found that in 2 mm deep cavities, using blocker resin composite with single-shade resin composite on dark-colored teeth had a darkening effect, causing minimal change in color. Similarly, applying blocker with group-shaded and multi-shaded resin composites on light-colored teeth showed the same effect with the light-colored teeth in 2 mm deep cavities in the visual evaluation by dentists. However, no difference in color matching was observed by the non-dentist group (*p* > 0.05). In the visual evaluation, statistically significant results were observed between dentists and non-dentists for both light- and darker-colored teeth when comparing the Estelite Sigma Quick without blocker on light-colored teeth (E-L; 0.46 ± 0.23) and Estelite Sigma Quick without blocker on dark-colored teeth (E-D; 1.76 ± 0.90). Among all groups, Estelite Sigma Quick without blocker on dark-colored teeth (E-D) exhibited the highest average value (Figure 2). These findings indicate that there are differences in color matching perception between dentists and non-dentists, particularly when using Estelite Sigma Quick without blocker on different shades of teeth.

## 4. Discussion

The first hypothesis of this study, which aimed to analyze the differences in color matching among single-shaded, group-shaded and multi-shaded resin composites, as well as the second hypothesis regarding the color matching of the tested composites with light and dark colored teeth, were rejected. The first and second hypotheses were rejected because single-shaded, group-shaded and multi-shaded resin composite systems did not achieve satisfactory color matching on both light and dark colored teeth in the instrumental evaluation. Specifically, single-shaded and group-shaded resin composites were unable to achieve harmonious color blending beyond the clinically acceptable level, while the use of multi-shaded resin composites with different color group compositions did not yield clinically acceptable results on light- and dark-colored teeth. The third hypothesis was rejected in terms of instrumental evaluation, as there was no observed effect of the blocker on the color matching between resin composites and natural teeth. However, in the visual evaluation, only dentists were able to discern the mismatch of single-shaded resin composite used with a blocker on darker-shaded teeth, as well as multi-shaded and group-shaded resin composites on light-shaded teeth. This result indicates that blocker application does not positively impact the color matching of composites with teeth (*p* < 0.05) for dentists alone, thus partially supporting the third hypothesis. However, as the effect of the blocker was not discerned by the non-dentist group, the third hypothesis was partially rejected.

Many factors, including the proportion and size of filling, matrix composition, size of restoration, the layering technique of the composite and color and brand of the composite, can affect the color matching of the tooth with resin composite [4,16]. It wasreported that resin composites containing smaller and irregularly shaped filler particles allow for better light transmission compared to those with larger particles [17]. Furthermore, they observed that irregularly shaped filler particles decrease the a * values and increase the b* values, in contrast to composites with spherical filler particles [17]. As composites include color pigments and metal oxides, the composition of the materials is also important. Titanium oxide increases the opaqueness of the composites and gives them an enamel-like look [18]. It was reported that if the size of the restoration and color difference decrease, the transparency increases BE [4]. In another study it was determined that BE is influenced by the shade and type of composite used [5].

Color matching is vital for the aesthetic restorations made with resin composites [19]. The acceptable perception range for the ΔE_00_ color parameter, according to CIEDE 2000, is between 0.8 and 1.8 at a 50–50% level. In a study in which ΔE_00_ was measured with a spectroradiometer on VITA ceramic samples, within a predetermined range of tooth colors, and it was classified as perceivable/acceptable by diverse observers [8]. However, another study conducted on acrylic teeth did not yield clinically acceptable results for multi-shaded, group-shaded and single-shaded resin composites [20]. Our instrumental evaluation results are consistent with a previous study on human incisor teeth, which also did not show clinically acceptable data for multi-shaded, group-shaded and single-shaded resin composites [20]. In our study, single-shaded composites exhibited the highest ΔE_00_ values in both groups with and without the blocker. In the instrumental analysis, group-shaded and multi-shaded composites demonstrated lower ΔE_00_ values for lighter- and darker-colored teeth. Additionally, the ΔE_00_ values for single-shaded resin composites were the highest compared to other groups. However, our study did not reveal significant color matching differences between multi-shaded and group-shaded composites for both light- and darker-colored teeth. None of the results yielded ΔE_00_ values below 1.8, which is considered clinically acceptable. This may be due to the polychromatic structure of natural teeth. Visual perception is subjective; however, it has a good adaptation mechanism that allows for a reasonable interpretation of different images to achieve a consistent color match [21].

A more practical method for color evaluation that was proven to be an effective parameter in earlier studies is visual grading [12]. In these earlier studies, it was found that multi-shaded composites achieved the best results on dark-colored teeth, while single-shaded and group-shaded composites yielded the best results on light-colored acrylic teeth [20]. In the present study, the multi-shaded composite in the E-D group obtained the highest values (1.49 ± 0.59), indicating clinically acceptable color matching to darker-colored human incisor teeth. The multi-shaded resin composite in the E-L group and the group-shaded resin composite in the NB-L group achieved lower grades ranging from 0.48 to 1.41, which indicated a full match and a very good match, respectively, to light-colored human incisor teeth.

“Color adjustment potential” (CAP) is used to distinguish physical and perceptual components of blending. CAP-I is used in the instrumental evaluation, whereas CAP-V is used for visual evaluation [3]. Single-shaded resin composite exhibited the best match on acrylic teeth in the visual evaluation [11]. The concept of “Color adjustment potential” (CAP) is used to differentiate between the physical and perceptual components of blending. CAP-I is utilized in instrumental evaluation, while CAP-V is employed in visual evaluation. Our results indicated that single-shaded resin composites demonstrated the best match on both light- and darker-colored human teeth. In the visual evaluation conducted by both dentists and non-dentists, group-shaded and multi-shaded composites showed clinically acceptable results for darker-colored teeth. Several factors influence the perception of color matching, including the morphology and location of the tooth to be restored on the dental arch, reflection of the oral cavity, the amount of remaining tooth tissue and the surrounding tissues around the teeth [20]. In addition, natural teeth possess a polychromatic, semi-transparent and curved morphology, which affects the reflection and distribution of light. All of these factors can indeed influence the behavior of resin composites in in vivo studies and their evaluation through instrumental methods. In our instrumental measurements of the effect of the blocker on single-shaded, group-shaded and multi-shaded composites applied to 2 mm deep cavities, we did not observe acceptable results. There was no statistical difference found in the instrumental evaluation and visual evaluation conducted by the non-dentist group among the teeth restored with multi-shaded, single-shaded and group-shaded resin composites, both with and without blocker application. However, the visual evaluation conducted by dentists indicated that the application of the blocker with single-shaded resin composites on darker-colored teeth or the application of group-shaded and multi-shaded resin composites on lighter-colored teeth negatively affected the color match. Although single-shaded composites did not achieve clinically acceptable results in the instrumental evaluation, they yielded clinically acceptable results in comparison to group-shaded and multi-shaded composites in the visual evaluation. Omnichroma, which is a pigment-free universal resin composite, contains uniformly spaced and organized spherical particles that allow for light transmission throughout the restoration. As a result of the restoration’s increased particle size and structure, the color of the restoration appears to match that of its surroundings [11].

Although the present in vitro study did not demonstrate a positive effect of blocker composite on 2 mm deep cavities, further research, both in vitro and in vivo, is necessary to examine the impact of blocker composite on deeper cavities. Additionally, it would be valuable to investigate the effect of the blocker on teeth with sclerotic dentin or teeth that require restoration with single-shaded, group-shaded and multi-shaded resin composites after the removal of amalgam restorations with dark-stained dentinal tubules. These investigations would provide a more comprehensive understanding of the potential benefits and limitations of blocker composite in different clinical scenarios.

## 5. Conclusions

Within the limitations of this study, the following results were obtained:The color matching is influenced by the composite material used, the color of the teeth and the remaining surrounding tissue.In the instrumental evaluation, single-shaded composites exhibited higher ΔE_00_ values compared to multi-shaded and group-shaded resin composites.In the visual evaluation, all observers found that single-shaded, group-shaded and multi-shaded composites achieved the best color matches, respectively.The application of blocker composite with single-shaded composites on dark-colored teeth resulted in minimal color changes in 2 mm deep cavities, as observed in the visual evaluation by dentists. Similarly, applying blocker with group-shaded and multi-shaded composites on light-colored teeth caused minimal color changes in 2 mm deep cavities, according to the visual evaluation by dentists.Overall, the application of blocker did not negatively affect the color matching of different shaded resin composites, as observed in the visual evaluation by all observers. This suggests that blocker can be used as a dentin shade in 2 mm deep cavities of anterior teeth.

## Figures and Tables

**Figure 1 materials-16-04954-f001:**
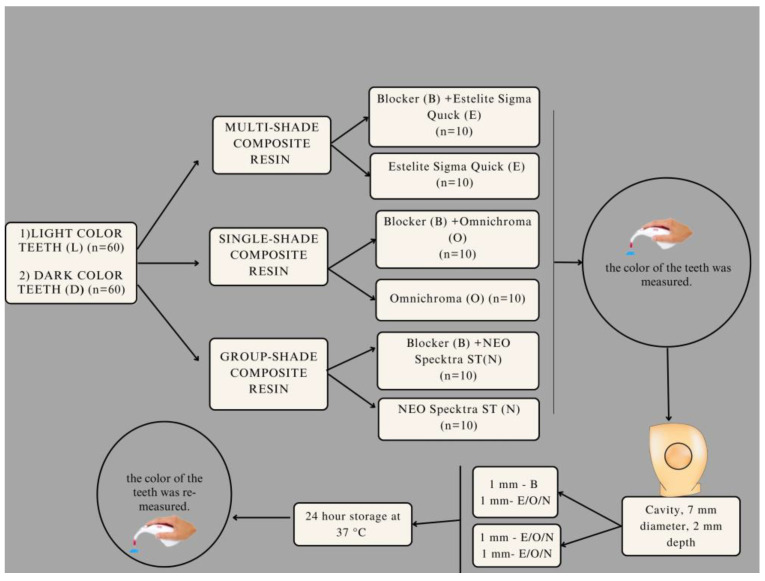
Schematic workflow of the study.

**Figure 2 materials-16-04954-f002:**
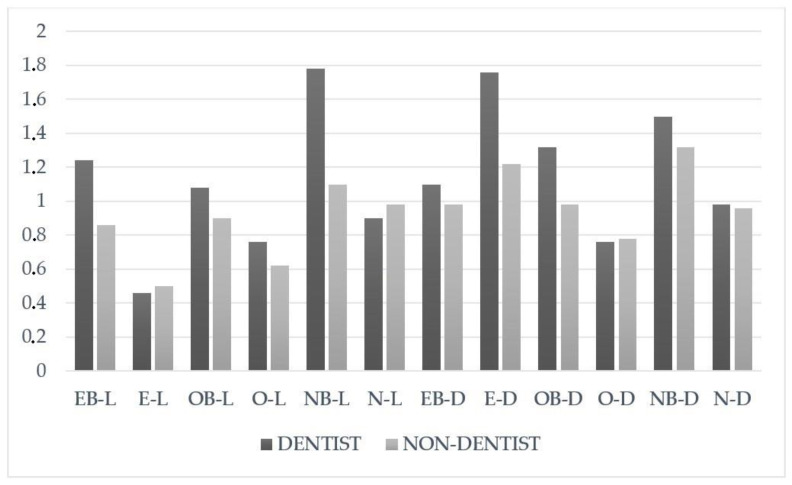
The comparison of visual evaluation averages of blocker status for tooth color and composite types evaluated by dentists and non-dentists. Abbreviations: “E” Estelite Sigma Quick, “O” Omnichroma, “N” Neo Spectra ST, “B”, Blocker, “L” light teeth, “D” dark teeth.

**Table 1 materials-16-04954-t001:** Composite materials evaluated in this study.

Material	Manufacturer	Filler Type	Filler Content wt% vol%	Monomer	Shades	Code
**Omnichroma (O)**	Tokuyama Dental, Tokyo, Japan	spherical silica zirconia filler and composite filler	82 71	UDMA, TEGDMA	Universal shade	0712
**Estelite Sigma Quick (E)**	Tokuyama Dental, Tokyo, Japan	supra-nano and spherical fillers	82 71	Bis-Gma TEGDMA	A1 A2 A3 A3.5 B1 B2 B3 B4 C1 C3	E3075 E8331 E0613 E0037 W69722 W6419 W7269 W51311 E077B2 W92421
**Neo Spectra ST (N)**	Dentsply Sirona Inc., Germany	Spherical, pre polymerized SphereTEC fillers and non-agglomerated barium glass and ytterbium fluoride	80 78	UDMA TEGMA BİSEMA	A1 A2 A3 A3.5	2110001020 2203000318 2202000346 2202000951
**Omnichroma Blocker (B)**	Tokuyama Dental, Tokyo, Japan	spherical silica zirconia filler and composite filler	82 71	Bis-GMA TEGDMA	-	35415

**Table 2 materials-16-04954-t002:** Comparison between groups with or without blocker in light- and dark-colored teeth in accordance with the type of resin composite used.

Group	ΔE_00_ Mean
**EB-L**	3.61 (1.72) ^a^
**OB-L**	5.46 (1.90) ^b^
**NB-L**	3.31 (1.14) ^a^
**EB-D**	3.38 (1.60) ^a^
**OB-D**	10.11 (0.74) ^b^
**NB-D**	3.11 (1.91) ^a^
**E-L**	3.61 (0.65) ^a^
**O-L**	6.51 (2.00) ^b^
**N-L**	2.64 (1.00) ^a^
**E-D**	3.79 (1.20) ^a^
**O-D**	10.52 (1.97) ^b^
**N-D**	3.61 (2.03) ^a^

Abbreviations: “E” Estelite Sigma Quick, “O” Omnichroma, “N” Neo Spectra ST, “B” Blocker, “L” light teeth, “D” dark teeth. Note: The use of the same upper symbol letters (“a” or “b”) indicates that there is no statistically significant difference between the composite materials within each group.

**Table 3 materials-16-04954-t003:** Average values for the visual evaluations by all observers.

Group	Visual Score _Mean_
**EB-L**	1.05 (0.73) ^a^
**OB-L**	0.99(0.49) ^a^
**NB-L**	1.44 (0.45) ^a^
**EB-D**	1.04 (0.68) ^a^
**OB-D**	1.15 (0.47) ^a^
**NB-D**	1.41 (0.51) ^a^
**E-L**	0.48 (0.28) ^a^
**O-L**	0.69 (0.35) ^a^
**N-L**	0.94 (0.47) ^a^
**E-D**	1.49 (0.59) ^a^
**O-D**	0.77 (0.38) ^b^
**N-D**	0.97 (0.52) ^a^

Abbreviations: “E” Estelite Sigma Quick, “O” Omnichroma, “N” Neo Spectra ST, “B”, Blocker, “L” light teeth, “D” dark teeth. Note: The use of the same upper symbol letters (“a” or “b”) indicates that there is no statistically significant difference between the composite materials within each group.

## Data Availability

Co-author Nuran Ulusoy has published the article below in ‘Pathogens’ in 2019. Karadağlıoğlu, Özgü İlkcan, Nuran Ulusoy, Kemal Hüsnü Can Başer, Azmi Hanoğlu, and İrem Şık. “Antibacterial activities of herbal toothpastes combined with essential oils against Streptococcus mutans.” Pathogens 8, no. 1 (2019): 20.

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
