# Peer review of "The Effect of Blocker Application on Color Matching of Different Colored Composite Resin Systems"

_materials, 2023, doi:10.3390/ma16144954_

Round 1

Reviewer 1 Report

This manuscript titled "The Effect of Blocker Application to Color Match of Different 2 Colored Composite Resin Systems" aims to evaluate and compare the effect of blocker application on the color match of light- and dark-colored natural teeth restored with single, group and multilayer composite resins. I believe the work is well structured with a well-formulated and organized design. The materials and methods are thoroughly explained and the paper is well written. 

In conclusion, I believe that the present work is interesting and can provide useful clinical implications to simplify patient management by dentists. Therefore, I believe the paper is potentially acceptable for publication, subject to some modifications that would enhance its quality. 

Figure 1: Improving the quality and organization of the same figure.

Minor editing of English language required

Author Response

Point 1: This manuscript titled "The Effect of Blocker Application to Color Match of Different 2 Colored Composite Resin Systems" aims to evaluate and compare the effect of blocker application on the color match of light- and dark-colored natural teeth restored with single, group and multilayer composite resins. I believe the work is well structured with a well-formulated and organized design. The materials and methods are thoroughly explained and the paper is well written. In conclusion, I believe that the present work is interesting and can provide useful clinical implications to simplify patient management by dentists. Therefore, I believe the paper is potentially acceptable for publication, subject to some modifications that would enhance its quality. 

Figure 1: Improving the quality and organization of the same figure.

Response 1: Thank you for your insightful comments, which we believe will be important for the manuscript. We have made changes to "Figure 1" according to your comments and we hope that the new figure meets your standards. The English language has also been edited.

Reviewer 2 Report

The article "The Effect of Blocker Application to Color Match of Different Colored Composite Resin Systems" evaluates and compare the effect of blocker application on color match of light and darker colored natural teeth restored with single, group and multi-shaded composite resins.

The article has serious major flaws that should be addressed before continuing review:

The English language needs complete proofreading.

It does not make sense to describe the type of composites in the introduction. It shall be done in the discussion.

No significant level (p=0.05?) was declared in the Statistical analysis.

No indications on how the teeth were initially stored were provided

Visual and Instrumental CAP (CAP-I and CAP-V) could have used to assess the color adjustment potential (article only partially cited in the Discussion. (CAP-I and CAP-V are optianl, but should be described in a proper paragraph).

Only 2 references were added in the discussion, and only one paper was compared to current study!

Huge English language proofreading is needed.

Author Response

Point1: It does not make sense to describe the type of composites in the introduction. It shall be done in the discussion.

Response 1: Thank you for your suggestion. We have moved and highlighted the paragraph you provided from the Introduction to the Discussion section:

"A lot of factors including the proportion and size of filling, matrix composition, size of restoration, the layering technique of the composite and color and brand of the composite can affect the color match of the tooth with resin composite [3,12]. Arikawa et al. (2007) reported that resin composites containing smaller and irregularly shaped filler particles allow for better light transmission compared to those with larger particles. Furthermore, they observed that irregularly shaped filler particles decrease the a* values and increase the b* values, in contrast to composites with spherical filler particles [13]. As composites include color pigments and metal oxides, the composition of the materials is also important. Titanium oxide increases the opaqueness of the composites and gives them an enamel like look [14]. Paravina et al. (2006) found that if the size of the restoration and color difference decrease; the transparency increases BE. In another study by Paravina et al. (2006), it was determined that BE is influenced by the shade and type of composite used."

Point 2: No significant level (p=0.05?) was declared in the Statistical analysis.

Response 2: Thank you for your comment. Statistical significance was accepted as p < 0.05 and added to statistical analysis section of the article.

Point 3: No indications on how the teeth were initially stored were provided

Response 3: Thank you for your comment. Prior to the commencement of the cavity preparation, the teeth were stored in distilled water. We have included the information in the material and method section of the article.

Point 4: Visual and Instrumental CAP (CAP-I and CAP-V) could have used to assess the color adjustment potential (article only partially cited in the Discussion. (CAP-I and CAP-V are optianl, but should be described in a proper paragraph).

Response 4: Thank you for your comment. We have added a new literature related with the below  content (Paravina RD, Westland S, Johnston WM, Powers JM. Color adjustment potential of resin composites. J Dent Res. 2008 May;87(5):499–503.) about CAP according to your suggestion: 'Color adjustment potential' (CAP), is used to distinguish physical and perceptual component of blending. CAP-I is used in the instrumental evaluation whereas CAP-V is used for visual evaluation. The concept of 'Color adjustment potential' (CAP) is used to differentiate between the physical and perceptual components of blending. CAP-I is utilized in instrumental evaluation, while CAP-V is employed in visual evaluation.'It would have been interesting to explore CAP-I which is used for assessing the translucency and CAP-V in the study. However, in the case of our study, its seems slightly out of the scobe. In the present study CAP-I and CAP-V evaluations were not used because we did not explore the translucency of the materials, we only evaluated the color match of resin composite with natural teeth.

Point 5: Only 2 references were added in the discussion and only one paper was compared to current study!

Response 5: Thank you for pointing this out. We agree that it is an important consideration but due to the novelty of the materials used in our study, the review section of our study had limited articles available for discussion. Since there is no study on the effect of blocker composite ussage with one-shaded, group-shaded and multi-shaded resin composites in the literature this study will be first research on blocker composite ussage for masking color changes. For this reason we could not find directly related literature to discuss.

Point 6: Huge English language proofreading is needed.

Response 6: Thank you for your comment. The English language has also been edited.

Reviewer 3 Report

Firstly, I would like to commend the authors on their research work, which addresses an important aspect of resin composite—the achievement of color match. The introduction provides a clear overview of the significance of color match in both aesthetic and functional aspects, emphasizing the challenges encountered when dealing with polychromatic natural teeth and the complexities involved in color selection and composite resin application. The authors have effectively highlighted the importance of this research area.

However, upon reviewing the literature review section, I believe it could benefit from a more comprehensive approach. While the authors have referenced relevant studies, the review primarily focuses on citing previous research without providing a critical analysis or synthesis of the existing literature. I recommend that the authors expand their review to include more recent studies and undertake a critical analysis to support the claims made in the introduction. This would enhance the credibility and relevance of the manuscript.

On the other hand, the discussion section of the research paper presents a comprehensive analysis of the study's findings and offers valuable insights. The inclusion of both instrumental and visual evaluations adds depth to the discussion and facilitates a meaningful comparison of the results. The identification of factors that can influence color match, such as tooth morphology and the presence of surrounding tissues, demonstrates the authors' understanding of the subject matter. Additionally, the mention of the impact of composite material composition, particle size, and structure on color match provides valuable and relevant information.

However, to strengthen the discussion, I suggest that the authors provide a more detailed explanation of why the hypotheses were rejected. Elaborating on the specific reasons for the rejection would enhance the readers' understanding and provide a clearer picture of the research outcomes. This additional information would further contribute to the overall quality of the manuscript.

Author Response

Point 1: However, upon reviewing the literature review section, I believe it could benefit from a more comprehensive approach. While the authors have referenced relevant studies, the review primarily focuses on citing previous research without providing a critical analysis or synthesis of the existing literature. I recommend that the authors expand their review to include more recent studies and undertake a critical analysis to support the claims made in the introduction. This would enhance the credibility and relevance of the manuscript.

Response 1: Thank you for your this suggestion. We have added the suggested content to the manuscripct and highlighted them in the indroduction section. We have also added a new literature related with the new content (Paravina RD, Westland S, Johnston WM, Powers JM. Color adjustment potential of resin composites. J Dent Res. 2008 May;87(5):499–503.)

Point 2: However, to strengthen the discussion, I suggest that the authors provide a more detailed explanation of why the hypotheses were rejected. Elaborating on the specific reasons for the rejection would enhance the readers' understanding and provide a clearer picture of the research outcomes. This additional information would further contribute to the overall quality of the manuscript.

Response 2: Thank you for your comment. We have added the suggested content to the manuscripct and highlighted it in the first paragraph of discussion.

Reviewer 4 Report

Overall, this is a very well planned and written study.  The results are interesting.

Abstract: The abstract is well written and easy to understand. However, the distribution of research groups should be described more precisely.

 It rads “Round cavities (7mm x 2mm) were prepared”. Please clarify how round cavities are measured (diameter? or?).

Keywords: simple for readers to understand and are relevant to the content.

Introduction: The introduction provides a good, generalized background of the topic.

Material and Methods:

This study has considered every important point required.

However, it is necessary to structure the text better, e.g. separating tooth preparation and color assessment separately.

Describe if and how the etching procedure was performed?

The ethical aspects of the study have not been addressed. You must mention the permission of the ethics commission and No.

Please provide a power analysis for your final sample size and group breakdown.

Results:

Results are difficult to understand, should be structured better.

It reads “Groups OB-L, OB-D, O-L and O-D”. Please explain the meaning of abbreviations in the text and next to the tables.

It reads “Same upper symbol letters show that there is no difference between the groups.” Please explain in more detail, can't understand the meaning.

Discussion: The discussion is well written.

Conclusions: The conclusions are well written.

Author Response

Point 1: The abstract is well written and easy to understand. However, the distribution of research groups should be described more precisely. It rads “Round cavities (7mm x 2mm) were prepared”. Please clarify how round cavities are measured (diameter? or?).

Response 1: Thank you for your this suggestion. We have explained the research groups again and clarified the diameter and depth of the cavities. We highlighted changes in the abstract.

Point 2: However, it is necessary to structure the text better, e.g. separating tooth preparation and color assessment separately.

Response 2: Thank you for your this suggestion. We agree with this and the preparation of the cavities were reorganised separately from the instermental and visual analysis. We changes were highlighted in the 2.1 and 2.2 of the material and method section.

Ponit 3: Describe if and how the etching procedure was performed?

Response 3: We applied ‘universal bond’; an one-step adhesive which combines acid, primer and bond together. We did not apply a separate etching procces.

Point 4: The ethical aspects of the study have not been addressed. You must mention the permission of the ethics commission and No.

Response 4: Thank you for pointing this out. Ethical approval (YDU/2022/108-1655) was granted by the Near East University Scientific Research Ethics Committee on November 30, 2022, ensuring compliance with ethical standards and participant protection. We changes were highlighted in the first paragraph of the material and method section.

Point 5: Please provide a power analysis for your final sample size and group breakdown.

Response 5: Thank you for pointing this out. A power analysis was conducted using analysis software (G*Power, Version "3.1.9.2") to determine the necessary sample size. The sample size for each group was determined to be a minimum of 5, with α = 0.05, 80% power and a 95% confidence level. To ensure the desired statistical power, ten samples were prepared for each test group. Power analyses are included in the statistical analysis in the article. We changes were highlighted in the first paragraph of the statistical analysis.

Point 6: Results are difficult to understand, should be structured better. It reads “Groups OB-L, OB-D, O-L and O-D”. Please explain the meaning of abbreviations in the text and next to the tables. It reads “Same upper symbol letters show that there is no difference between the groups.” Please explain in more detail, can't understand the meaning.

Response 6: Thank you for your comment. We have added the suggested content to the manuscripct and highlighted them in the results, tables and figures. The groups with blockers were labeled with the letter 'B', the single shade composite Omnichroma was labeled with the letter 'O', the group shade composite Neo Spectra ST was labeled with the letter 'N' and the multi-shade composite Estelite Sigma Quick was labeled with the letter 'E'. Light shade teeth were denoted by the letter 'L' and dark shade teeth were denoted by the letter 'D'. Within each group, comparisons were made. The groups with blockers were compared among themselves and the groups without blockers were compared among themselves. The superscript 'a' indicates no statistical difference, while 'b' indicates a statistical difference.

Round 2

Reviewer 2 Report

All comments and requests have been amended.

Reviewer 4 Report

Accept in present form